# Mixed-methods feasibility study to inform a randomised controlled trial of proton pump inhibitors to reduce strictures following neonatal surgery for oesophageal atresia

Tracy Karen Mitchell [1], Nigel J Hall [2], Iain Yardley [3,4], Christina Cole [5] Pollyanna Hardy [5], Andy King [5], David Murray [5], Elizabeth Nuthall [5], Charles Roehr [5], Kayleigh Stanbury [5], Rachel Williams [5], John Pearce,[6] Kerry Woolfall [1]

For numbered affiliations see end of article.

**Correspondence to**
Dr Tracy Karen Mitchell;
Tracy.Mitchell@Liverpool.ac.uk

## ABSTRACT

**Objectives** This mixed-methods feasibility study aimed to explore parents' and medical practitioners' views on the acceptability and design of a clinical trial to determine whether routine prophylactic proton pump inhibitors (PPI) reduce the incidence of anastomotic stricture in infants with oesophageal atresia (OA).

**Design** Semi-structured interviews with UK parents of an infant with OA and an online survey, telephone interviews and focus groups with clinicians. Data were analysed using reflexive thematic analysis and descriptive statistics.

**Participants and setting** We interviewed 18 parents of infants with OA. Fifty-one clinicians (49 surgeons, 2 neonatologists) from 20/25 (80%) units involved in OA repair completed an online survey and 10 took part in 1 of 2 focus groups. Interviews were conducted with two clinicians whose survey responses indicated they had concerns about the trial.

**Outcome Measures** Parents and clinicians ranked the same top four outcomes ('Severity of anastomotic stricture', 'Incidence of anastomotic stricture', 'Need for treatment of reflux' and 'Presence of symptoms of reflux') as important to measure for the proposed trial.

**Results** All parents and most clinicians found the use, dose and duration of omeprazole as the intervention medication, and the placebo control, as acceptable. Parents stated they would hypothetically consent to their child's participation in the trial. Concerns of a few parents and clinicians about infants suffering with symptomatic reflux, and the impact of this for study retention, appeared to be alleviated through the symptomatic reflux treatment pathway. Hesitant clinician views appeared to change through discussion of parental support for the study and by highlighting existing research that questions current practice of PPI treatment.

**Conclusions** Our findings indicate that parents and most clinicians view the proposed Treating Oesophageal Atresia with prophylactic proton pump inhibitors to prevent STricture (TOAST) trial to be feasible and acceptable so long as infants can be given PPI if clinicians deem it clinically necessary. This insight into parent and clinician

## STRENGTHS AND LIMITATIONS OF THIS STUDY

⇒ A mixed-methods approach including a survey, interviews and focus groups enabled comprehensive insight into key stakeholder views.
⇒ Despite the difficulties experienced in arranging interviews, we continued to interview parents until the point of information power and to involve parents of infants with oesophageal atresia (OA) at all stages, including study design and conduct, as members of the study team.
⇒ Our sample may comprise experienced parents with an interest in OA research and may not reflect the potential Treating Oesophageal Atresia with prophylactic proton pump inhibitors to prevent STricture sample who will also have less awareness of proton pump inhibitor and treatment options for symptoms of reflux at the time the trial is discussed.
⇒ Our study includes the perspectives of clinicians involved in the treatment of OA representing the majority of UK surgical units.

views and concerns will inform pilot phase trial monitoring, staff training and the development of the trial protocol.

## INTRODUCTION

Oesophageal atresia (OA) is a rare congenital anomaly that affects a baby's oesophagus, where the upper part of the oesophagus does not connect with the lower part. As this is life-threatening, surgery is usually carried out shortly after birth. Approximately 150 babies are born with OA annually in the UK.[1] Stricture (narrowing) at the anastomosis (new connection) is the most common postoperative complication in the months following surgical repair,[1,2] which requires admission to

hospital for investigation and dilatation of the narrowed segment under general anaesthesia.[2]

Some international guidelines[3] recommend routine proton pump inhibitors (PPIs) for all infants with OA for the first year of life to reduce the incidence of anastomotic stricture; currently, just over 50% of surgeons in the UK prescribe PPI prophylactically to babies with OA.[1] Babies are then managed by surgeons and neonatologists following hospital discharge. Some studies[1 4 5] and a systematic review and meta-analysis of the evidence,[6] however, indicate that infants routinely given PPI are no less likely to get a stricture. The evidence to support the use of PPI is not conclusive and stems from only a small number of low quality, observational or single-centre studies.[6] Furthermore, PPI can increase the risk of gastrointestinal[1 5 7 8] and respiratory infections,[3 7] raising concerns about giving medication to infants that has no benefit.

A randomised controlled trial is needed to answer the question 'In infants born with OA, does the routine use of PPI compared with matched placebo impact the incidence or severity of anastomotic stricture?' The chances of successful trial completion are improved if the trial is deemed to be acceptable to parents and clinicians. This paper presents the findings of a mixed-methods feasibility study, which aimed to explore parent and practitioner views on the feasibility, acceptability and design of a proposed randomised controlled trial: Treating Oesophageal Atresia with prophylactic proton pump inhibitors to prevent STricture (TOAST).

## METHODS

### Study design

We conducted a mixed-methods study involving interviews (June–September 2021) with parents of an infant born with OA in the last 3 years, as well as an online survey (August–October 2021), interviews (October 2021) and focus groups (November–December 2021) with clinicians caring for infants with OA.

We used previous research[9 10] to develop participant information sheets (PIS) (see online supplemental file 1), protocol and online survey (see online supplemental file 2), while ongoing findings were used to develop topic guides (see online supplemental file 3) and as part of an iterative process. Topic guides and the survey included questions on the proposed trial design, information materials, trial acceptability, willingness to be involved/ provide consent, the approach to consent and parent prioritised outcomes for the proposed trial. The research was conducted in the UK between June and December 2021. The Consolidated criteria for Reporting Qualitative research checklist[11] was used to aid reporting (see online supplemental file 4).

### Patient and public involvement

Our parent advisory group (PAG) involved members of TOFS charity, who support infants born with OA/TOF (tracheo-oesophageal fistula). The PAG met regularly before and during the study, providing valuable input into the design of research materials (including topic guide and draft reflux treatment pathway) and the conduct, progress and findings of this study. JP (TOFS Trustee) was a member of the TOAST study management team and a TOFS representative for all aspects of study development and conduct.

### Recruitment and sampling procedure

Based on previous feasibility studies,[9 10 12] we anticipated that we would need to interview 15–25 parents to reach information power,[13] which is the point at which data addresses the study aims; sample specificity (e.g., participants' experience relevant to the study aims and sample diversity);[11] our reflexive and interpretive approach to theory and analysis[14 15]; and sufficient quality of interview dialogue.[13] Parents were recruited via direct email from our collaborating support group TOFS, as well as via social media and study website advertising.

We aimed to recruit at least 50 clinicians to the online survey from approximately 18/25 (75%) of UK units. IY (male, paediatric surgeon) distributed an invitation to participate in the survey through the UK Children's Upper Gastrointestinal Surgery (ChUGS) network with a request to cascade the survey link to clinicians involved in the care of OA infants. We aimed to purposively sample clinicians who raised concerns about the proposed trial design in their survey responses and invite them to participate in a telephone interview to further explore their concerns and discuss potential ways these could be addressed to assist 'buy in'. Finally, we invited survey participants to attend an online or face-to-face focus group.

### Eligibility screening and conduct

TKM (female, research methodologist) responded to parents' email and social media responses in sequential order, confirmed eligibility and emailed them a proposed trial Parent Information Leaflet (PIL) (see online supplemental file 5), draft treatment pathway for symptomatic reflux (see online supplemental file 6), and potential list of outcome measures (see online supplemental file 7) derived from a review of the literature. KW (female, social scientist) contacted clinicians to arrange interviews and IY sent invitations to attend a focus group. TKM and KW facilitated interviews and focus groups. Respondent validation was used to add unanticipated topics to the topic guide as interviewing and analysis progressed.[16] Findings from parent interviews and online survey were used to develop the topic guide for the clinician interviews and focus group. Interviews stopped when information power[13] was achieved and all clinicians who responded to the invite were interviewed. Parents were sent a £30 shopping voucher after their interview to thank them for their time.

**Table 1** Parent and child characteristics

| Parent | Mother (n=13) / Father (n=5) |
|---|---|
| Parent age | Between 29 and 42 years (mean=36 years; median=36 years) |
| Child age | Between 4 weeks and 34 months (mean=14.1 months old; median=17 months) |
| Gestation | Term (n=14) |
| | Premature (n=3) $31^{+0}$ weeks, $33^{+0}$ weeks and $33^{+3}$ weeks |
| Country of residence | England (n=15) |
| | Scotland (n=3) |
| Ethnic group | White British (n=15) |
| | White Scottish (n=1) |
| | White other (n=1) |
| | Indian (n=1) |

**Table 2** Clinician characteristics

| Method of data generation | No of clinicians and role | No of sites represented |
|---|---|---|
| Online survey | n=51: 49 paediatric surgeons and 2 neonatologists | 20 (80%) |
| Interview | n=2: consultant paediatric surgeons | 2 (8%) |
| Focus group | n=10 Focus group 1: 5 consultant paediatric surgeons. Focus group 2: 2 consultant paediatric surgeons, 2 paediatric surgeons and 1 consultant neonatologist. | 9 (36%) |

## Analysis

TKM led the analysis with oversight from KW. Analysis of direct questioning and indirect discussion was broadly interpretive and inductive, informed by the theoretical framework of acceptability (TFA) and adapted version for paediatrics.[9 17] NVivo V.12 software[18] was used to assist the organisation and coding of data. TKM and KW met regularly to discuss interpretation and develop the coding framework. Outcome measures prioritised as being most important were given a score of 3, second most important a score of 2 and third most important a score of 1. Outcomes were then ranked. Quantitative data were entered into Microsoft Excel.[19] Descriptive statistics are presented with frequencies and percentages. Synthesis of qualitative and quantitative data for mapping findings to the TFA drew on the constant comparative method.[20 21]

## RESULTS
## Sample

A total of 39 parents registered interest and were screened. Three parents were deemed ineligible, 3 booked interviews but cancelled due to their child's hospital readmission and 15 parents did not respond to initial contact. Information power was reached at 18 parent interviews (representing 17 children), which took place via telephone (n=15) or online (n=3), lasting between 40 and 92.5 min, mean 63.6 min, median 65 min (see table 1). Nine parents were recruited through TOFS, three from social media (Facebook) and six could not recollect whether TOFS email or Facebook.

Fifty-one clinicians (49 paediatric surgeons; 2 neonatologists) from 20/25 (80%) sites completed the online survey. Four of the six clinicians (paediatric surgeons) who indicated in the survey that they did not find the trial acceptable had provided their contact details and were contacted to take part in an interview. Two did

not respond to contact and two surgeons from different sites took part in an online Zoom interview (lasting 23 and 27 min). Ten clinicians from 9 different sites (36%) took part in 1 of 2 focus groups, one face to face (n=5 surgeons), 1 online via Zoom (n=4 surgeons; n=1 neonatologist). Both focus groups lasted 1 hour. See table 2 for clinician characteristics.

## Trial research question

Across the research methods, the majority of parents and clinicians (through survey responses) described or indicated that the proposed trial would answer an important research question and help address *'how little evidence there is'* (P10, mother, interview). Some parents spoke of their hope that the study would help future babies with OA, whilst both parents and clinicians stated the trial was needed to help standardise practice and prevent babies from taking potentially unnecessary medication, while also reducing costs and burden for families and the NHS.

> I think it [the proposed trial] is a good thing, because I hear a lot of parents on the TOFS site and things, and they are obviously getting different medical care and their concerns about that really. I think it is something that needs to be standardised (P11, mother, interview).

> If significant difference found then potential to decrease burden on families and providers (C44, surgeon, survey).

> If your child doesn't need to be on a medication, then you don't really want them to be on it (P13, mother, interview).

## Parent information

Interviews and focus groups involved a review of a draft trial PIL. The majority of parents said that they found the proposed PIL to be clear and understandable. However, some stated it was *'quite long'* (P10, mother, interview) and *'text heavy'* (P16, father, interview), while acknowledging all necessary information was included.

Recommended changes included adding a one-page overview of the study; highlighting the differences in treatment that are already happening and not using acronyms.

## Symptomatic reflux treatment pathway

The study team recognised the need to develop a tool to assist clinicians in making decisions about how to treat babies who had symptoms of reflux during the proposed trial. A symptomatic reflux treatment pathway was developed, which included options for non-pharmacological treatments (e.g., exclude overfeeding, keep baby upright after feeds) and time frames for re-evaluation (e.g., every 2weeks). During interviews and focus groups, the draft symptomatic reflux treatment pathway was described as *'helpful for parents and clinicians'* (C51, surgeon, survey). Parents' suggestions for improvement were mainly around the additional symptoms of reflux, signs of stricture, other non-pharmacological treatments that could be initiated and accessibility of the document (see online supplemental file 8a). Clinicians suggested adding conditions such as tracheomalacia (C22, surgeon, survey), the timing of/whether babies have *'anti-reflux'* surgery (C27, surgeon, survey) and prioritising breastfeeding over formula feeding (C41, neonatologist, survey) (see online supplemental file 8b).

Although the treatment pathway had been originally designed for use by clinicians during the trial, parents highlighted how it would be helpful to refer to and *'be aware of the things that are written down… just as a reminder of, for example, it says, 'Are they gaining weight adequately? … Is he crying normally?'* (P7, mother, interview). One father suggested that the pathway will *'make them [parents in the trial] feel more comfortable'* (P8, interview) about trial participation. Parents said they would be happy to follow the symptomatic reflux treatment pathway, *'so long as no child is being left to suffer'* (P13, mother, interview) and '*the health of the individual child would trump… being in the study*' (P18, father, interview).

Most clinicians (n=38/51, 74.5%) indicated in the survey that they would be happy to follow the treatment pathway (see online supplemental file 6); others raised concerns about the potential 4-week time frame to PPI (n=5); the severity of reflux symptoms (n=4) and retention of participants (n=3):

Slight concern that it may be difficult to get TOFOA parents to agree to … wait a further 4 weeks … if their child is symptomatic (C49, surgeon, survey)

This would be fine for minor symptoms but inappropriate for severe symptoms (C12, surgeon, survey)

I think if there is a clinician who wants to take someone out of it [the trial], you could use that escalation policy [symptomatic reflux treatment pathway] to do so…That's the difficult thing, I think (C29, surgeon, interview).

## Support for omeprazole as the intervention, but some concerns about side effects

Most clinicians (including 60.8% of survey participants) routinely administered or prescribed PPI following surgery in all babies with type C OA under their care. During interviews and focus groups, some clinicians stated that they did not prescribe PPI following surgery due to a lack of evidence about stricture formation, or when patients did not have any symptoms of reflux. Side effects of PPI, such as the increased risk of infections, not knowing the long-term risks or wanting to minimise unnecessary drug use, were also reasons not to use PPI.

Nevertheless, all parents and the vast majority of clinicians, found omeprazole acceptable as the trial intervention as it was a *'routinely used by many teams with a very safe profile'* (C35, surgeon, survey). The dose of 1mg/kg omeprazole orally once daily for 1year was also described as being acceptable, although four mothers and two clinicians perceived 1mg/kg omeprazole per day to be a low dose, and had *'slight concern that it may be difficult to get parents [of children with OA] to agree to that dose'* (C49, surgeon, survey). Some parents, however, wondered whether PPI *'actually had any effect'* (P17, father, interview) because their child *'still had a stricture'* (P12, mother, interview) even though they had taken PPI from birth, while one father (P1, interview) and one surgeon (C21, focus group 1) said that children were left off PPI and *'nothing happened anyway'* (C21, surgeon, focus group 1). A minority of parents and clinicians had concerns about side effects, such as *'it [omeprazole] seemed to thicken her mucus a lot, so it produced more blue episodes'* (P16, father, interview), *'a very sore tummy'* (P9, mother, interview) and *'sepsis/G.I. infections'* (C40, surgeon, survey), and were concerned about the trial length due to the long-term impacts of the medication. One surgeon said that infants should take PPI for *'6 months only to avoid side effects'* (C26, survey).

## Treating reflux in the comparator arm and the challenge of changing practice

The use of a placebo in the comparator arm of the proposed trial was acceptable to both groups, although parents stated that babies should not be *'left to suffer'* with reflux (P13, mother, interview) and the symptomatic reflux treatment pathway should *'be implemented sensibly'* (P4, mother, interview), however, clinicians should not *'automatically assume you need'* PPI (P9, mother, interview). A minority of clinicians were concerned about a change in practice and placing babies at risk of negative outcomes if they were not given PPI in the trial, particularly if they have symptomatic reflux and tight anastomosis:

Babies in the placebo group are exposed to a high risk of complications… It is not safe to have a baby post-TOF without PPI (C9, surgeon, survey).

I would struggle to join a clinical trial where I know that there is a randomisation of my symptoms who were not using PPIs… I was taught the importance

**Table 3** Proposed inclusion and exclusion criteria

| Inclusion criteria | Exclusion criteria |
|---|---|
| ► Infants with OA with distal tracheo-oesophageal fistula undergoing primary repair at the first operative intervention in the newborn period<br>► Written informed parental consent | ► Infants undergoing staged repair or delayed primary repair or requiring emergency ligation of tracheo-oesophageal fistula with primary repair later<br>► No realistic prospect of survival |

OA, oesophageal atresia.

of the PPIs [during my career] and I think it make sense to use PPIs in this condition [OA]…if you do any repair of a tissue, you don't want to spill acid on it (C9, surgeon, interview).

### Inclusion and exclusion criteria
Most clinicians who took part in the survey and focus groups were satisfied with the proposed inclusion and exclusion criteria (see table 3). Recommendations for improvement covered five key areas: (1) Include babies who require 'delayed' or 'staged' repairs (n=11) *'as PPI might be beneficial in those'* (C43, surgeon, survey); (2) Exclude babies with tight anastomosis because of the perceived increased risk of *'reflux'* (C23, surgeon, survey) and *'stricture formation'* (C3, surgeon, survey) (n=7); 3) Exclude preterm babies - *'Use of PPI in preterm is not neutral, and has been shown to be associated with NEC (necrotising enterocolitis) and fungal in sepsis'* (C33, neonatologist, survey) (n=5); (4) Exclude babies with other anomalies (n=9) such as *'coexistent duodenal atresia or ARM'* (C45, surgeon, survey), *'cardiac/renal/neurological/chromosomal'* (C7, surgeon, survey), *'congenital oesophageal stenosis (COS)'* (C27, surgeon, survey), *'HIE/major brain injury … and VACTERL (vertebral defects, anal atresia, cardiac defects, TOF, renal anomalies and limb abnormalities)'* (C4, surgeon, survey) and (5) Include but consider *'the homogeneity of the… population'* of (C27, surgeon, survey) babies who have thoracoscopic rather than open repairs (n=2).

### The importance of not discussing the trial on the day of surgery
Parents were then asked to consider when would be the most acceptable time to be approached about the proposed trial. Most stated that 2–3 days after birth would be best, as long as they have received *'the good news of [their baby having] a successful repair'* (P18, father, interview) and when their baby is *'starting to look stable'* (P4, mother, interview) and is off the ventilator. A clear message from parents was that trial discussions on the day of surgery would be too overwhelming. Some suggested the trial could be discussed with parents prior to birth if OA is diagnosed antenatally. Clinicians made similar recommendations to broach the discussion within 72 hours post-surgery. The concerns of the three clinicians who did not find it acceptable to approach parents at this time

were, once again, around the safety of infants with OA who do not receive PPI:

> I usually start PPI from the time of surgery. It is not acceptable to leave the baby without PPI for 72 hours. The baby should be randomised before the surgery (C9, surgeon, survey).

### The use of a mobile application to assist trial retention
Parents' views were sought on the use of a mobile phone application (app), which would include reminders to administer the trial intervention when they had left hospital. Most parents thought that the app was a good idea and would be a *'massive bonus'* (P7, mother, interview) and so *'useful'* (P13, mother, interview) *'to offer with'* the trial (P18, father, interview). Seven parents felt that the app is not needed, although were not averse to having an app for the trial, so long as it would not be a mandatory requirement for parents to use it.

Furthermore, when questioned about content that might be useful in an app, most made a number of suggestions such as: *'hints'* (P12, mother, interview), *'tips'* (P6, mother, interview) *'and advice … on how to [prepare and] administer'* (P13, mother, interview) the intervention; reminder notifications; symptoms tracker and a medical history page because *'the days all merge… [and] sometimes I'll be like, 'Oh yes, he's been coughing.' And then the surgeon will be like, 'So how long has that been going on for?' And I'm like, 'Oh Gosh, I don't know'* (P3, mother, interview). Other suggestions included information about the study and the main signs of reflux and stricture and *'a guide to CPR because I know a lot of parents are very, very anxious about that'* (P4, mother, interview).

### Shared views on outcomes of importance
Parents and clinicians were then asked to consider a list of potential outcomes sent prior to interview and focus groups, as well as any additional outcomes they felt should be included. Parents and clinicians suggested edits or additions to most of the predefined outcomes, as shown in online supplemental file 9.

Participants were asked to rank the outcomes that were most, second and third most important to be measured in the TOAST trial. After weighting, 'severity of anastomotic stricture', 'incidence of anastomotic stricture', 'need for treatment of reflux' and the 'presence of symptoms of reflux' remained the four most important outcome measures for the TOAST trial for both parents and clinicians.

### Potential barriers to trial success
In the survey, 38 (38/51; 74.5%) clinicians stated they were a little (n=32) or very (n=6) concerned about the retention of babies in the trial if they have symptomatic reflux and were receiving the placebo, reflecting the concerns of some parents and clinicians interviewed:

> You might put a baby in a placebo group, but if they have reflux, they'll have to have the antacid

medication, and reflux is really- well, as far as I'm aware, really common (P3, mother, interview).

If babies become symptomatic then parents may ask to come out of trial and be assured that they're on a PPI (C20, surgeon, survey).

Some clinicians' concerns about retention of participants in the trial related to stricture management in babies with signs of reflux:

If a patient has a particularly difficult stricture, and signs of reflux I would want to know if they are being treated or just on placebo, as at this point, I would definitely want them on a PPI (C15, surgeon, survey).

Other clinicians' concerns were about geographical challenges, highlighting the need for *'as many continuity sites as possible'* (C33, surgeon, survey) to be tertiary centres and how *'some of our remote/poorer patients would struggle to travel to face-to-face follow-up'* (C4, surgeon, survey). A combination of external factors and trial setup queries were discussed including: the *'differing views of… surgical and neonatal (and other) colleagues'* (C21, surgeon, survey) about preference for use of PPI and prescribing outside of the trial; quality of intervention blinding and sourcing; staffing and research support issues, especially *'out of hours'* (C2; C30, surgeons, survey); access to training and support needs; and, reflecting the concerns of a small number of parents, the pro-medication influence of TOFS Facebook group members:

It will be interesting to see what the parents have said, and what, like the TOFS group says, because I think most of the parents will be members of that group, and what they feel about reflux and how willing they would be if they go on the forum and say, "Oh, I think my kid's refluxing and he's on this trial. What should I do?" What advice they're going to be given from the parent groups because I think that would be a big factor (C29, surgeon, interview).

It was only when I joined the TOFS Facebook group that I thought, "Oh dear there's a lot of stuff going on and a lot of complications and a lot of people talking about medication the whole time" (P9, mother, interview).

### Overall views on trial acceptability

Towards the end of the interview, survey or focus group participants were asked to consider the overall acceptability of the trial. All 18 parents stated that the proposed trial was acceptable; 3 with the proviso that their child could access PPI medication if clinically necessary and so long as *'a nice, softly-softly approach'* (P16, father, interview) was taken by an experienced and *'trusted doctor or surgeon'* (P9, mother, interview). Having trust in the opinions of health professionals about their child's involvement in the trial was mentioned (unprompted) by over half of parents.

**Table 4** Adapted [*9] theoretical framework of acceptability[17].

| Construct | Definition |
|---|---|
| Affective attitude | How an individual feels about the intervention |
| Burden | The perceived amount of effort that is required to participate in the intervention |
| Ethicality | The extent to which the intervention has a good fit with an individual's value system |
| Intervention coherence | The extent to which the participant understands the intervention and how it works |
| Opportunity costs | The extent to which benefits, profits or values must be given up to engage in the intervention |
| Perceived effectiveness | The extent to which the intervention is perceived likely to achieve its purpose |
| Self-efficacy | The participant's confidence that they can perform the behaviour(s) required to participate in the intervention |
| Trust* | The extent to which the participant (or parent/guardian) trusts those delivering the intervention to put the needs of patient before the requirements of the study |

*, the addition of Trust[9] to the original Theoretical Framework of Acceptability[17].

Almost all clinicians stated that they found the proposed trial to be acceptable overall, despite the potential barriers to success described above. The views of the two clinicians who found the trial 'not acceptable' in the survey appeared to shift in favour of the trial during their subsequent interview, during which the evidence which questioned the use of omeprazole was discussed and changes to the reflux treatment pathway were explained, including parent views.

Finally, our findings were considered against the adapted TFA for paediatric trials (p. 9)[9], (p. 522)[17], which consists of eight component constructs (see table 4).

Analysis of feasibility study data indicates that five out of eight constructs of the TFA (affective attitude, burden, intervention coherence, self-efficacy and trust) for the TOAST trial were fully met for parents. Concerns of a minority related to the ethicality construct and the proposed omeprazole dose (1 mg/kg) being insufficient to treat reflux symptoms and potential side effects. The remaining constructs were largely met, or could be met, if suggestions for changes to the trial materials and protocol are addressed by the team.

Although almost all clinicians stated they found the proposed TOAST trial acceptable overall, only two out of seven constructs of the TFA (affective attitude and burden) were fully met for clinicians who completed the survey and three met (affective attitude, burden and opportunity costs) for those who took part in the focus group or interviews. As the themes presented in this paper highlight, wider issues impacted on anticipated acceptability including: the ability to retain patients in the

trial due to concerns about a potential 4-week time frame to PPI for babies with symptomatic reflux; a change in practice and the need to amend the inclusion criteria to make the trial more acceptable for some.

## DISCUSSION

This study provides insight into the acceptability of the proposed TOAST trial for parents and clinicians who care for infants with OA. Like other studies that highlight the value of feasibility work[10] and 'conducting pretrial research with key stakeholders' (p. 9)[9] to improve recruitment and retention in clinical trials,[22 23] involving parents and clinicians in this feasibility study provided valuable insight into potential barriers and solutions to recruitment and retention of infants in the TOAST trial.

Overall, the majority of parents and clinicians who took part in this feasibility study supported the proposed trial as they felt it would help address an area of clinical uncertainty. Parents and clinicians ranked the same top four outcomes ('severity of anastomotic stricture', 'incidence of anastomotic stricture', 'need for treatment of reflux' and the 'presence of symptoms of reflux') as important to measure for this study. Our findings highlight the need to carefully consider how symptomatic reflux would be treated in all trial participants. Although all parents found the use, dose and duration of omeprazole as the intervention medication and placebo control acceptable, some parents whose child had experienced signs of symptomatic reflux[8] had concerns about being able to access PPI if their child was in discomfort. Parents of children who had experienced commonly reported side effects of PPI, such as infections, wind or an upset stomach,[1 3 5 7 8] or a previously unreported side effects, such as thick mucus that made breathing difficult, stated they would still hypothetically consent for their child to take part in the trial even if they had a 50/50 chance of receiving PPI.

Our findings show that despite clinicians stating that they found the trial acceptable, multiple constructs in the TFA were not fully met due to concerns or perceived challenges to conducting the trial. Some were external factors that they felt parents may face, such as the ability of families to travel to follow-up appointments, or the pro-medication influence of TOFS Facebook group impacting on the views of new parents of OA infants invited to participate. As also found by others,[24] most other challenges raised related to changing usual individual clinical practice, and for this study, clinician equipoise and specifically a wish to access PPI when children were showing signs of reflux, in line with existing guidelines.[3] As described above, these findings echo the concerns of some parents. Refining the inclusion and exclusion criteria and developing a symptomatic reflux pathway that clinicians would find acceptable will be key to ensuring they are willing to enrol infants in their care into the trial. While reviewing this pathway with parents during interviews, it became apparent that they also viewed the pathway as an important resource for parents in the trial, which may

assist with participant retention. Many felt it would bring reassurance that babies 'would not be left to suffer' with symptoms of reflux if they took part in the TOAST trial. Parents supported the use of an 'opt in' mobile phone application that would send reminders to administer the trial intervention, as well as host the symptomatic reflux pathway and other related trial information, all of which may help with protocol adherence and help prevent withdrawal from the trial.

Previous research has shown the importance of identifying when is an appropriate time to discuss trial participation, as a poorly timed approach can cause additional burden for distressed families, which may also increase the likelihood that parents will decline trial participation.[25 26] Parents stated that it would not be acceptable for clinicians to broach the trial on the day of surgery as it would be too overwhelming. Time points before surgery, when a baby is diagnosed with OA during pregnancy, and 2–3 days after surgery were recommended by both parents and clinicians.

We found that the views of a minority of clinicians whose survey responses suggested they did not find the trial acceptable appeared to shift in favour of the trial during subsequent interviews. During these conversations, the evidence that indicated PPIs may increase stricture rates,[5 6] parent views and the proposed reflux treatment pathway were explained. This finding alone highlights content that should be included in staff training and trial resources, as well as wider findings that demonstrate parental support for the trial. Inclusion of a statement on the treatment pathway which states that if clinicians 'feel that urgent treatment is needed, clinical judgement takes precedence' is also likely to help address concerns about the potential 4-week time frame to administer PPI, and therefore, make the trial seem more acceptable. However, it is also important to recognise that such a statement may also lead to cross-over between trial arms, or patient withdrawal, which should be closely monitored in the pilot trial phase.

### Strengths and limitations of this study

A mixed-methods approach including a survey, interviews and focus groups enabled comprehensive insight into key stakeholder views, as well as the ability to explore clinician concerns that were evident in the survey in more depth through interviews. Although 39 parents registered interest in an interview, nearly half did not respond to further correspondence and three cancelled due to their child being readmitted to hospital, which highlights the challenges of engaging parents of such vulnerable children in research. Despite the difficulties experienced in arranging interviews, we continued to interview parents until the point of information power and our study included parents with recent relevant experience. As the majority of parents were recruited through the TOFS support group, our sample may comprise experienced parents with an interest in OA research and may not reflect the potential TOAST sample who will also have

less awareness of PPI and treatment options for symptoms of reflux at the time the trial is discussed. The clinicians involved were overwhelmingly surgeons, with only two neonatologists taking part in the survey. However, as surgeons will predominantly be deciding which babies to approach for the TOAST trial, this is unlikely to a be significant limitation to assessing the feasibility of the trial.

## CONCLUSIONS

All parents and most clinicians viewed the proposed trial as being feasible and acceptable, so long as infants can access PPI if clinically required. Our findings will inform the trial protocol for the internal pilot phase of the main trial as well as the main trial itself and site training materials to ensure the trial is family centred and to assist clinician engagement. Recruitment, retention and protocol adherence data should be closely monitored during the pilot phase to inform decisions about progression to a full trial.

**Author affiliations**

[1]Department of Public Health, Policy and Systems, Faculty of Health and Life Sciences, Institute of Population Health, University of Liverpool, Liverpool, UK
[2]University Surgery Unit, Faculty of Medicine, University of Southampton, Southampton, UK
[3]Evelina Children's Hospital, Guy's & St. Thomas's NHS Foundation Trust, London, UK
[4]Faculty of Life Sciences and Medicine, King's College, London, UK
[5]National Perinatal Epidemiology Unit, Clinical Trials Unit, Nuffield Department of Population Health, University of Oxford, Oxford, UK
[6]TOFS, Nottingham, UK

**Acknowledgements** Special thanks are extended to TOFS parent advisory group members for their involvement in the design, conduct, progress and findings of the feasibility study and the development and conduct of the proposed trial. Gratitude is extended to Alan Downs[5] for his involvement in this study.

**Contributors** TKM, NJH, IY, CC, PH, AK, DM, CR, KS, RW, JP and KW were involved in the conception or design of the study; or the acquisition, analysis or interpretation of data for the work. TKM and KW collected the data. TKM analysed and interpreted the data with oversight from KW. TKM and KW drafted the article and NJH, IY, CC, AK, EN, CR, KS, RW and JP critically revised it for important intellectual content. TKM, NJH, IY, CC, PH, AK, DM, EN, CR, KS, RW, JP and KW gave approval of the version to be published. TKM and KW agree to act as guarantors and to be accountable for all aspects of the work in ensuring that questions related to the accuracy or integrity of any part of the work are appropriately investigated and resolved. TKM, KW, NJH and IY revised the article after peer review.

**Funding** The TOAST feasibility study was funded by the National Institute for Health Research Health Technology Assessment Programme (project number NIHR 131136; link to Protocol: https://fundingawards.nihr.ac.uk/award/NIHR131136) and coordinated by the University of Liverpool.

**Disclaimer** The views and opinions expressed herein are those of the authors and do not necessarily reflect those of the HTA Programme, NIHR, NHS or the Department of Health. The funders had no role in the collection, analysis and interpretation of data or in the decision to submit this article for publication.

**Competing interests** National Institute for Health Research (NIHR) Health Technology Assessment (HTA) Programme grant payments to support the conduct of this study were made to the institutions of all authors (except JP). TKM and EN employment roles were funded by the NIHR HTA programme grant payment made to their institutions.

**Patient and public involvement** Patients and/or the public were involved in the design, or conduct, or reporting, or dissemination plans of this research. Refer to the Methods section for further details.

**Patient consent for publication** Consent obtained directly from patient(s).

**Ethics approval** This study involves human participants and was approved by University of Liverpool Research Ethics Committee. Approval reference number 8510. Participants gave informed consent to participate in the study before taking part.

**Provenance and peer review** Not commissioned; externally peer reviewed.

**Data availability statement** No data are available. Although data relevant to the study are included in the article and are uploaded as online supplemental information, no raw data are available. The datasets generated and analysed for this study are not publicly available due to participants being from a small, specialised population, which creates ethical and privacy concerns about being too identifiable.

**ORCID iDs**
Tracy Karen Mitchell http://orcid.org/0000-0003-0014-8016
Nigel J Hall http://orcid.org/0000-0001-8570-9374
Iain Yardley http://orcid.org/0000-0002-6928-2267
Christina Cole http://orcid.org/0000-0002-8798-2136
Pollyanna Hardy http://orcid.org/0000-0003-2937-8368
Andy King http://orcid.org/0000-0001-5180-7179
David Murray http://orcid.org/0000-0001-9010-2905
Elizabeth Nuthall http://orcid.org/0000-0002-5092-7643
Charles Roehr http://orcid.org/0000-0001-7965-4637
Kayleigh Stanbury http://orcid.org/0000-0002-8726-2411
Rachel Williams http://orcid.org/0000-0002-5872-1690
Kerry Woolfall http://orcid.org/0000-0002-5726-5304

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
