## [Reviewer comments · BMJ Open]

ARTICLE DETAILS

TITLE (PROVISIONAL)	A mixed methods feasibility study to inform a randomised controlled trial of proton pump inhibitors to reduce strictures following neonatal surgery for oesophageal atresia
AUTHORS	Mitchell, Tracy; Hall, Nigel; Yardley, Iain; Cole, Christina; Hardy, Pollyanna; King, Andy; Murray, David; Nuthall, Elizabeth; Roehr, Charles; Stanbury, Kayleigh; Williams, Rachel; Pearce, John; Woolfall, Kerry

VERSION 1 – REVIEW

REVIEWER	Walk, Ryan Uniformed Services University of the Health Sciences F Edward Hebert School of Medicine, Surgery
REVIEW RETURNED	10-Oct-2022

GENERAL COMMENTS	Congratulations to the authors for their work. For all of the reasons outlined in the paper, the cost-benefit of PPIs after repair of esophageal atresia remains a crucial unanswered question in the care of these challenging patients. This paper presents the authors initial steps towards answering the question via a randomized trial. Their mixed-methods feasibility study provides a fascinating window into the concerns raised by major stakeholders (mainly parents and surgeons) over their proposed study's design. As the authors acknowledge, their samples of both parents and clinicians are subject to bias. The parents were recruited from online support groups and a charity foundation, and the participating parents skew older than would otherwise be expected. These factors give the impression of having recruited from an affluent and more highly educated cohort; generalizing their concerns may not be appropriate. The clinicians are overwhelmingly surgeons. As the provided quotations from the qualitative interviews illustrate, several hold a dogmatic attachment to the use of these medications post-operatively. Some background in regards to the management of these medications in British centers would be helpful. In the United States, while there will be some variability between centers, for the most part, neonatologists initiate the medications post-operatively and gastroenterologists manage them after discharge. The surgeons' concerns highlighted in the paper would not provide as significant a hurdle to implementation. The exclusion criteria proposed by the participating clinicians provides an enlightening summary of the authors' challenge. In
---

	aggregate, these would exclude all but a small number of the most straightforward cases: in theory, at least, those least likely to benefit from the medication. Nevertheless, the question is an important one, and, as the participants' comments highlight, the stakes are quite high. This study provides an initial step in the right direction.
--	---

REVIEWER	Kunisaki, Shaun M Johns Hopkins Children's Center
REVIEW RETURNED	15-Oct-2022

GENERAL COMMENTS	This is a survey/mixed methods study evaluating the feasibility of a randomized trial of PPI use in the first year of life following esophageal atresia repair. The paper is well-written and logical. Below are my minor concerns: 1) Has a statistical power analysis been performed to determine how many enrolled patients would be required to see an effect with PPI vs. no PPI. I have concerns that this study would take 5-10 years to complete if limited to UK participants. If there is lack of power, then the survey might need to be opened to a wider audience 2) Can the authors comment on those under high tension (eg type A) and whether they would be excluded. What about those who have Nissen funduplications of nasoduodenal tubes during infancy? 3) The other possible benefit of PPI has to do with esophagitis and long-term cancer surveillance. Can the authors comment on how this would be mentioned in the trial? 2)
--

REVIEWER	Donald, GraemeDonald, Graeme The University of Manchester
REVIEW RETURNED	18-Nov-2022

GENERAL COMMENTS	Thank you for submitting this paper, which I enjoyed reading. I have been asked to focus my review on methods and analyses. I wouldn't call this a feasibility study - to me, it is more developmental, the stage before a feasibility study. Your methods assess anticipated acceptability from parents and clinicians but a feasibility study should be some kind of experimental study that produces data on lived, rather than anticipated, acceptability. The outcomes of a true experimental feasibility study should include, for example, recruitment rates, missing OM data items, adherence, attrition, and OM data that can inform a power calculation for a main trial and indicate the likelihood of clinically/statistically significant OM changes in a fully powered trial. I note that you mention progressing to a pilot after this which makes me wonder if this isn't simply an issue of terminology - I would call this a developmental study, as per eliciting input from key stakeholders to finalise the nature of the intervention and proposed study processes, whilst feasibility and pilot studies are both options for the design of the next stage of the research programme, which is also necessary before a full trial, even if it is simply because of the need for data to conduct a power analysis to predict trial sample size. All pilots are feasibility studies but not all feasibility studies are pilots. Please review publications of the last few years on the design of feasibility studies and the adapted CONSORT statement for reporting pilot and feasibility studies. I would like to see this paper published once it is presented
--

differently,

VERSION 1 – AUTHOR RESPONSE

Reviewer 1 Some background in regards to the management of these medications in British centers would be helpful. In the United States, while there will be some variability between centers, for the most part, neonatologists initiate the medications post-operatively and gastroenterologists manage them after discharge. The surgeons' concerns highlighted in the paper would not provide as significant a hurdle to implementation.	A sentence ' Currently, just over 50% of surgeons in the UK prescribe PPI prophylactically to babies with OA [1]. Babies are then managed by surgeons and neonatologists following hospital discharge ' has been added to the second paragraph of the introduction.
Methods	
Reviewer 2 Has a statistical power analysis been performed to determine how many enrolled patients would be required to see an effect with PPI vs. no PPI. I have concerns that this study would take 5-10 years to complete if limited to UK participants. If there is lack of power, then the survey might need to be opened to a wider audience.	A statistical power analysis been performed for the full trial although this detail is beyond the scope of this paper. The trial protocol will be published in due course.
Reviewer 2 Can the authors comment on those under high tension (e.g. type A) and whether they would be excluded. What about those who have Nissen funduplications of nasoduodenal tubes during infancy?	In the inclusion and exclusion criteria section of the paper (page 8) a high tension (tight anastomosis) was suggested by a survey participant. Nissen funduplications of nasoduodenal tubes was not discussed by participants in our sample. Our findings have been used to develop the trial protocol, including the inclusion and exclusion criteria. This will be submitted for publication in the near future.
Reviewer 2 The other possible benefit of PPI has to do with esophagitis and long-term cancer surveillance. Can the authors comment on how this would be mentioned in the trial?	Thank you for your comment. This is beyond the scope of this particular manuscript. Only one participant in our sample mentioned cancer.
Reviewer 3 I wouldn't call this a feasibility study - to me, it is more developmental, the stage before a feasibility study. Your methods assess anticipated acceptability from parents and clinicians but a feasibility study should be some kind of experimental study that produces data on lived, rather than anticipated, acceptability. The outcomes of a true experimental feasibility study should include, for example, recruitment rates, missing OM data items, adherence, attrition, and OM data that can inform a power calculation for a main trial and indicate the likelihood of clinically/statistically significant OM changes in a fully powered trial. I note that you mention progressing to a pilot after this which makes me wonder if this isn't simply an issue of terminology - I would call this a developmental	For feasibility studies that include a pilot trial we agree that these data would be expected. This feasibility study did not include a pilot trial as the funder (who defined it as a feasibility study) identified the need for qualitative research in the first instance to help ensure the design would be acceptable to families. Feasibility studies can have different designs, including a more qualitative focus on acceptability to help prepare for a larger, more definitive piece of research. A pilot study may have the same aim but a more experimental design (small RCT) which we believe the reviewer is referring to. TOAST will have an internal pilot with stop go criteria to answer some of the questions the

study, as per eliciting input from key stakeholders to finalise the nature of the intervention and proposed study processes, whilst feasibility and pilot studies are both options for the design of the next stage of the research programme, which is also necessary before a full trial, even if it is simply because of the need for data to conduct a power analysis to predict trial sample size. All pilots are feasibility studies but not all feasibility studies are pilots.	reviewer raises. We agree that it is a difference in terminology.
Reviewer 3 Please review publications of the last few years on the design of feasibility studies and the adapted CONSORT statement for reporting pilot and feasibility studies.	Thank you for your comment. The CONSORT statement relates to randomised pilot and feasibility studies and as we did not include randomisation in our feasibility study this is not relevant. However, to assist in the transparency and quality of reporting for this qualitative study we have completed the COREQ checklist which is uploaded as an Appendix (Supplementary file 4).